# Improving Balance and Movement Control in Fencing Using IoT and Real-Time Sensorial Feedback [note 1]

**DOI:** 10.3390/s23249801

**Published:** 2023-12-13

**Authors:** Valentin-Adrian Niță, Petra Magyar

**Affiliations:** 1Department of Communications, University Politehnica of Timișoara, 300006 Timișoara, Romania; 2Department of Physical and Sports Education, West University of Timisoara, 300223 Timișoara, Romania; petra.magyar94@e-uvt.ro

**Keywords:** sensorial feedback, real-time monitor, IoT, balance and movement control in fencing

## Abstract

Fencing, a sport emphasizing the equilibrium and movement control of participants, forms the focal point of inquiry in the current study. The research endeavors to assess the efficacy of a novel system designed for real-time monitoring of fencers’ balance and movement control, augmented by modules incorporating visual feedback and haptic feedback, to ascertain its potential for performance enhancement. Over a span of five weeks, three distinct groups, each comprising ten fencers, underwent specific training: a control group, a cohort utilizing the system with a visual real-time feedback module, and a cohort using the system with a haptic real-time feedback module. Positive outcomes were observed across all three groups, a typical occurrence following a 5-week training regimen. However, noteworthy advancements were particularly discerned in the second group, reaching approximately 15%. In contrast, the improvements in the remaining two groups were below 5%. Statistical analyses employing the Wilcoxon signed-rank test for repeated measures were applied to assess the significance of the results. Significance was solely ascertained for the second group, underscoring the efficacy of the system integrated with visual real-time feedback in yielding statistically noteworthy performance enhancements.

## 1. Introduction

The sports landscape is transforming through technology integration, particularly with the increasing prevalence of wearable devices utilizing inertial measurement units (IMUs). The extant body of research substantiates the efficacy of these devices in enhancing the training experience [1,2,3,4,5,6,7,8]. Traditionally positioned on the hip [9], alternative placements such as the wrist [10,11], thigh [12], knee [13], or even the back [14] are viable options contingent upon the specific demands of a given sport. Notably, across a spectrum of sports like tennis [15], football [16], basketball [17], handball [18], hockey [19], and martial arts [20], insufficient attention has been directed towards fencing in the existing research literature.

Fencing, characterized by its occurrence on a specialized surface measuring 14 m in length and 1.5 m in width, akin to a chessboard for two participants, places significance on the positional dynamics between adversaries. Irrespective of the opponent’s physical attributes, celerity, or strength, effective counteraction hinges upon the judicious manipulation of distance and positioning. To optimize such elements, fencers must cultivate speed, agility, force, endurance, and an acute sense of balance and movement control through specific training. While extensive literature exists for the enhancement of speed and agility in general [21,22,23,24] and within the domain of fencing [25], the assessment and improvement of hand control [26,27], there is a conspicuous absence of tools or devices for measuring balance and movement control. Usually, IMUs are used for assessing injury risks, as in [9,10,11,12] and [14], while the proposed system is used to evaluate balance and movement control performance. It is essential to mention that the proposed system considers the particular limitations of the movements made in fencing.

Current coaching practices rely heavily on visual assessment by experienced coaches, which is particularly challenging when dealing with a large cohort of athletes—perhaps 100 fencers in training and 10 to 20 in competition. Unfortunately, the traditional approach necessitates a sequential evaluation, limiting the feasibility of continuous assessment. In addressing this gap, a pioneering Internet of Things (IoT) system has been developed and successfully tested for automated real-time measurement of balance and movement control [28]. This innovative system empowers coaches to conduct comprehensive and instantaneous evaluations of balance and movement control for all their fencers, presenting a transformative solution to the existing challenges in the field. In his book, *This is Fencing!*, Ziemowit Wojciechowski (one of the world’s most renowned and sought-after foil coaches with a long and illustrious record of success) speaks about the importance of performance analysis, which can be “qualitatively based on observations or quantitatively based on factual or statistical data” [29].

Figure 1 illustrates a simplified representation of the motion dynamics in a fencing game. As depicted, the primary motion predominantly occurs along the X-axis. In this context, it is crucial to note that the torso’s angular velocity along the Z-axis is generally close to zero, with exceptions occurring during specific actions like counterattacks and close encounters. Additionally, any rotation around the X-axis by the fencer leads to undesirable side imbalance and is a behavior that should be avoided in all circumstances.

The sole permissible rotation, albeit limited in magnitude, occurs along the Y-axis. This is due to the unique leg movement involved in fencing: when moving forward, the front leg is raised and advanced, followed by the back leg, resulting in a slight rearward tilt. Conversely, the back foot is repositioned before the front leg, causing a subtle forward tilt when moving backward. Because of this specific footwork, a fencer’s torso should ideally maintain a nearly constant zero angular velocity around the X and Z axes. Moreover, if the movement is executed with proper balance control, the angular velocity around the Y-axis should be kept to a minimum. A professional fencer’s movement should closely resemble a train on tracks, with smooth back-and-forth motion and minimal tilting, ideally exhibiting angular velocities of 0 along all three axes (X, Y, and Z).

The angular velocity of the torso can be monitored in time using a gyroscope worn on the back of the fencer.

This current study examines an expanded iteration of the system introduced in prior work [28], aiming to enhance its functionalities beyond measuring and monitoring balance and movement control. The extended system is designed to actively improve fencers’ capacities by incorporating real-time haptic and visual feedback mechanisms. The experimental design involves three distinct groups of fencers: a control group comprising 10 participants who will undergo training without the utilization of the proposed system, another group of 10 fencers who will engage with the system incorporating visual feedback, and a third group of 10 fencers who will interact with the system comprising haptic feedback. This intervention will span five weeks.

## 2. Materials and Methods

Figure 2 delineates the primary components integrated into the proposed system. At the core of this system is the “Gyroscope sensor”, a pivotal element employed for real-time monitoring of fencers’ balance. This is achieved by analyzing angular velocity along the fencer’s torso’s X, Y, and Z axes. The “Balance and movement control monitor” is an Android application used by fencers and coaches for comprehensive performance tracking over temporal intervals. The “Haptic feedback” module is embodied by a specialized smartwatch designed explicitly for fencing [30]. This device emits vibrations if the gyroscope sensor detects imbalances surpassing a predefined threshold. This threshold is adjustable per the fencers’ anticipated performance levels, categorized based on their proficiency levels—ranging from beginner to professional. Concurrently, the “Visual feedback” mechanism manifests as a device featuring a colored LED signaling system. The LED emits a light green hue when the fencer’s movements align with acceptable balance standards corresponding to their skill levels. Conversely, an orange signal is activated if the fencer’s performance falls below the designated reference level, calibrated in accordance with their experience.

The “Audio feedback” component, still in the developmental phase, incorporates the utilization of headphones. It is designed to emit a distinct auditory cue when fencers exhibit suboptimal balance and movement control levels. It is imperative to note that this article focuses on the analysis of the “Visual feedback” and “Haptic features”, while the “Audio feedback” feature remains under active development.

### 2.1. Balance and Movement Control Sensor

The fundamental constituent within the envisaged sensor is a gyroscope, a device instrumental in the measurement and preservation of orientation and angular velocity. Consisting of a rotating wheel or rotor affixed to gimbals, this apparatus facilitates unimpeded rotation in all directions. Upon the initiation of rotor motion, its axis of rotation remains steadfast, unaffected by the device’s movements.

The gyroscope’s functioning is grounded in the conservation of angular momentum. As the rotor spins, it possesses a fixed amount of angular momentum, resisting alterations in its orientation. Consequently, if the device is rotated, the rotor maintains its original orientation, causing the gimbals to revolve around it.

Figure 3 depicts the hardware components of the balance sensor, including an 1100 mAh battery, a Wemos Lolin 32 Lite (an Arduino-type board) equipped with Wi-Fi capabilities, and an MPU6500 sensor with a 3-axis gyroscope (offering programmable full-scale ranges of ±250, ±500, ±1000, and ±2000 degrees per second, and minimal noise at 0.01 degrees per second per square root of Hertz), an accelerometer, and a digital motion processor. The accelerometer features user-programmable full-scale ranges of ±2 g, ±4 g, ±8 g, and ±16 g. Initial sensitivity calibration for both sensors minimizes production-line calibration needs. This device is designed to operate in temperatures ranging from −40 °C to 80 °C and with voltages between 1.71 V and 3.6 V. With Wi-Fi continuously active, it typically consumes an average of 150 mA, providing about 7 h of autonomy with a 1100 mAh battery. This translates to approximately three fencing training sessions before requiring recharging. The total estimated cost of the components is roughly $15, with an additional $5 for a 3D-printed enclosure (as shown in Figure 4) and a strap for wearing the sensor on the back, positioned between the shoulders.

Recognizing the niche nature of fencing as a sport and the limited demand for such a device, the enclosure (Figure 3) has been designed for 3D printing per order. It has dimensions of 60 mm by 60 mm and a height of 20 mm, featuring a screwless design for easy manual assembly. The design is based on the model available at [31]. The total 3D printing time is approximately 3 h, requiring around 30 g of PLA filament. The final enclosure weighs less than 100 g and can be comfortably worn with a back strap without hindering a fencer’s performance on the fencing piste.

The equilibrium sensor depicted in Figure 3 establishes connectivity with the internet through Wi-Fi, facilitating data storage in a cloud-based database. This configuration allows fencers to monitor their performance metrics about balance contemporaneously. Authorization to access this data is contingent upon individual fencer preauthorization, and further permissions can be granted to share this information with their respective coaches. This collaborative feature furnishes coaches with a comprehensive overview of the progress exhibited by all athletes under their tutelage.

Data access is facilitated for both fencers and coaches, subject to explicit permission granted by the fencers, and is executed through a server infrastructure. Data processing occurs on the server, while the smartphone application retains a transient copy of the data. This temporary copy is automatically deleted if fencers revoke access privileges for a specific coach.

Two discrete cohorts of fencers were meticulously selected to establish benchmarks utilizing the data from the intelligent balance sensor. The initial group, encompassing novice practitioners, comprised ten individuals with fewer than twelve months of fencing experience. Conversely, the second group, comprised of proficient fencers, consisted of ten individuals with a notable 4 to 5 years of fencing practice. The observation of these fencers transpired during their traversal along a 14 m fencing strip, both at 50% of their maximum speed and at full acceleration, denoted as 100% speed, as detailed in a prior study [28].

To conduct a performance evaluation between novice and experienced fencers, a meticulous measurement of their angular velocity along the X, Y, and Z axes was undertaken, employing a time resolution of 100 milliseconds. The resultant data were aggregated for each fencer by applying the mean absolute deviation (MAD).

MAD, as described in [32], is a statistical metric for assessing the average deviation between individual data points and the mean of the dataset. Its calculation involves determining the absolute difference between each data point and the dataset’s mean, summing these fundamental differences, and dividing the total by the number of data points in the set.

The mean absolute deviation (MAD) is expressed in the same units as the original dataset, which measures how dispersed the data are from their mean. A higher MAD value in our datasets indicates that a fencer exhibits unsteady or imbalanced movement, while a lower MAD signifies a fencer with advanced fencing skills. We chose MAD as a preferable alternative to the standard deviation, another commonly used measure of data spread. Unlike standard deviation, MAD is less affected by outliers, making it particularly valuable when extreme values, such as those caused by a contra attack resulting in high angular velocity on the Z-axis, might distort standard deviation calculations.

Each fencer’s movement control and balance performance were quantified using a trio of values: MAD of angular velocity on the X, Y, and Z axes. Subsequently, an unsupervised machine learning algorithm, K-means, was employed to partition the results of the two groups and evaluate whether these results could effectively differentiate between novice and advanced fencers.

K-means clustering, a well-established unsupervised machine learning algorithm [33], categorizes and segments data into clusters based on their similarities. This algorithm divides data points into k clusters with a centroid or central point. The algorithm initiates with random centroids and assigns data points to the nearest centroid. Centroids are then updated to reflect the mean of the given data points. This process repeats until convergence, achieved when centroids no longer change significantly or when a maximum number of iterations is reached.

Figure 5 displays the results at 50% speed, while Figure 6 presents the results at 100% speed. These figures are two-dimensional, considering only the MAD of angular velocity on the X and Y axes, though clustering also employs the Z-axis data. The automatic separation into two clusters remains consistent in both scenarios, demonstrating the reliability of the monitored data for distinguishing between experienced and inexperienced fencers. The centroids obtained can serve as reference points to identify fencers making progress in their movement control and those who may benefit from additional preparation.

Furthermore, these data enable coaches to spot outliers, such as highly talented fencers who could be primed for high-performance training or those with consistently poor results, who may be better suited for recreational fencing rather than pursuing international medals. It is important to note that these proposed indicators rely solely on balance and movement control and should not be the sole performance metrics. They can be complemented by indicators of speed, reaction time, and precision abilities to enhance the selection of fencers with potential for high performance [28].

### 2.2. Haptic Feedback Module

The haptic module is in the form of a smartwatch developed specifically for fencing [30].

In Figure 7, the schematic representation illustrates the configuration of the haptic module integrated into the proposed system. This module is designed to furnish real-time feedback to fencers in the event of heightened imbalance, as expounded upon in greater detail in [30], which provides an exhaustive delineation of its features. For the immediate context, two primary functions are harnessed: the vibration motor, constituting the haptic feedback mechanism directed towards the fencers, and the Wi-Fi capabilities, essential for the reception of real-time commands from the balance sensor. At the core of this module resides an Arduino-based board, specifically the NodeMCU, as depicted in Figure 7. This board encapsulates a Wi-Fi ESP ESP8266 microchip, confined within a compact two by three cm rectangle, incurring an approximate cost of five euros.

Figure 8 illustrates the schematic representation of the enclosure design for the haptic module.

### 2.3. Visual Feedback Module

The visual feedback module facilitates interaction with the balance sensor by utilizing the identical NodeMCU employed in the haptic feedback module. This NodeMCU is intricately linked to an 8.6 cm WS2812 RGB LED ring. The WS2812 delineates a smart LED light source family characterized by integrating the control circuit and the RGB chip within a compact 5050-packaged unit. Figure 9 depicts the configuration of the visual feedback module. The LED light is strategically positioned at one extremity of the fencing strip, ensuring its perpetual visibility to the fencer undergoing training. In this scenario, fencers are necessitated to maintain continuous attention on the LED, mirroring the visual vigilance imperative in a fencing bout where constant visual analysis of the opponent is required. The LED emits a light green hue when the fencer’s movements align with acceptable balance standards corresponding to their skill levels. Conversely, an orange signal is activated if the fencer’s performance falls below the designated reference level, calibrated in accordance with their experience.

### 2.4. Population and Sample

This investigation engaged juvenile athletes from the ACS Floreta Fencing Club in Timișoara aged 11 to 14. Explicit written authorization from the parents or legal guardians of the athletes was secured to facilitate their involvement in the research. To augment the authenticity and scholarly import of the study, a meticulous selection procedure was implemented, adhering to precisely delineated inclusion and exclusion criteria. These criteria were systematically classified into two distinct groups to ensure transparency, as outlined below:Inclusion criteria:
Subjects must be between 11 and 14 years old at the time of selection.They should have 4–5 years of experience in fencing.They must have the written consent of their parents/legal guardians for their participation in the study.Exclusion criteria:
Unmotivated absence from training sessions (not more than 4 times/5 weeks) and tests.Excused absence from more than four training sessions during the study. This kind of absence can be encountered in the context of competitions, training camps, school exams, or the occurrence of some illnesses.


The athletes were divided into three distinct groups: the control group (CG), the experimental group utilizing the visual feedback module (VG), and the experimental group employing the haptic feedback module (HG). Homogeneity across the groups was maintained concerning age, gender, and training proficiency, and the allocation process was executed through randomization, employing a lot-drawing method for both female and male participants.

Each group adhered to a regimen of four training sessions per week, wherein 30 min per session was designated explicitly for targeted exercises aimed at improving movement control. Notably, all groups engaged in identical exercises, with the sole divergence being that the VG and HG groups had access to the system incorporating the feedback module.

The evaluative benchmark comprises a structured 4-2-2-4 scenario using three strategically positioned poles at varying distances. These poles are situated at the commencement line, 2 m from the starting line, and 4 m from the starting line, respectively. Fencers are tasked with traversing the prescribed course, involving movement from the start point to the 4 m pole and back, followed by a sequence of movements from the starting point to the 2 m pole and back repeated twice. Subsequently, they navigate once more from the start point to the 4 m pole and back. Cumulatively, this entails forward and backward movements of 12 m each, encompassing seven alterations in movement direction. Comprehensive assessments were conducted on all fencers at the initiation of the study and subsequently repeated after a 5-week interval from the commencement of the investigation.

The angular velocity during movement is compared with reference thresholds. Determination of reference thresholds emanates from the cluster’s centroid associated with advanced fencers, as depicted in Figure 6, and is delineated in Table 1. A performance falling below the reference threshold indicates commendable results, while a performance surpassing the threshold is deemed undesirable. To quantify improvements in balance and movement control throughout the study, the temporal aspect has been encapsulated by measuring the time percentage. This metric represents the proportion of time relative to the entirety of the benchmark test, during which fencers perform above the reference thresholds. To mitigate the potential impact of measurement errors throughout the comprehensive evaluation, an over-threshold performance is defined when at least two thresholds are exceeded or when all three thresholds are concurrently surpassed.

The comparative analysis between the initial and final test results entailed a relative average comparison within each group. Additionally, the Wilcoxon signed-rank test for repeated measures was employed to ascertain the statistical significance of improvements within each group. Furthermore, the inter-group improvements were subject to scrutiny through the Kruskal-Wallis [34] test, applied to the difference vectors derived from the initial and final measurements. Pairwise comparisons between groups, specifically CG vs. VG, CG vs. HG, and VG vs. HG, were conducted using the Mann-Whitney U test [35], with Bonferroni-corrected *p*-values [36]. Notably, non-parametric tests were selected for these analyses due to the limited sample size. The objective was to monitor the statistical significance of the outcomes, considering the inherent constraints associated with the modest number of samples.

## 3. Results

### 3.1. Average Absolute Evaluation of the Efficacy of the Proposed Training Feedback Modules

In the Appendix A, we can see the change in the performance of the evaluated groups in terms of total movement time and unbalanced time in Appendix A. Furthermore, Table 2 and Table 3 serve as central repositories for critical outcomes, facilitating a comparative analysis of data from both absolute and relative perspectives. Across all three groups, marginal enhancements are discerned concerning the total movement time, indicative of a negligible evolution in the speed of the fencers over the 5-week testing interval. Upon scrutiny of the balance time, nominal progress is observed within the control group, a marginal improvement in the cohort subjected to the haptic feedback module, and a notable 18% amelioration in unbalanced time for the group utilizing the visual feedback module for training. This improvement is measured in the context of unbalance detected on all three axes and a 15.5% improvement when considering unbalance on two of the three. However, despite an overall improvement in the average, individualized examination of results, as delineated in the Appendix A, reveals instances where certain fencers exhibit deterioration in both movement and balance control. Figure 10, Figure 11 and Figure 12 provide a visual means for the comparative assessment of overall average results for the Control Group (CG), the Visual Feedback Group (VG), and the Haptic Feedback Group (HG).

### 3.2. Relative Evaluation of the Efficacy of the Proposed Training Feedback Modules Using the Wilcoxon Signed-Rank Test for Repeated Measures

When evaluating group performances based solely on the overall absolute average, a potential limitation exists wherein a minority of subjects may demonstrate substantial improvements. At the same time, the majority may experience marginal enhancements or, in some instances, a decline in performance. Relying solely on the average might suggest significant improvement attributable to a particular method, yet random cases could influence such observations. To ascertain the genuine impact of the proposed modules on enhancing balance and movement control in fencing, a secondary assessment of the results is conducted using the Wilcoxon signed-rank test for repeated measures [37]. The detailed outcomes of this test, comparing unbalance time on 2 out of 3 axes and 3 out of 3 axes for CG, VG, and HG, are presented in Appendix A.

Table 4 shows the overall results of the Wilcoxon test applied to all three test groups. Two overall results are of fundamental importance: the Wilcoxon test result and the critical value. If the test result is under the critical value, the improvements in balance and movement control are statistically significant; if it is not, they are not statistically significant. On the first line, we can see the results for the control group; in both situations, unbalanced time on two out of three axes and unbalanced time on three out of three axes, we can see that the Wilcoxon test points out that the improvements are not statistically significant. Even when we analyze Table 3, minor improvements are obtained from an average point of view. The same thing results for the group that has been training using the proposed system with haptic feedback. Real improvements were obtained from the group that was trained using the system with a visual feedback system.

### 3.3. Vector Difference Comparison of the Final and Initial Testing between the Three Groups Using the Kruskal–Wallis and Mann–Whitney U Tests, with Bonferroni-Corrected p-Values

An examination of the outcomes depicted in Figure 10, Figure 11 and Figure 12 reveals discernible improvements in the performance of all groups following five weeks of training, specifically regarding movement control and balance. However, a meticulous inspection of Table 4 discerns that statistically significant improvements, at the 5% significance level, are exclusively evident for the group that availed the visual feedback module. In this subsection, we aim to test two hypotheses:Each of the three groups has been drawn from a population with an identical distribution.Following the five-week training period, the enhancements realized by one group demonstrate statistical significance when compared to the other groups within the study.

Examining the first hypothesis will involve the comparison of performance pairs, namely CG vs. VG, CG vs. HG, and VG vs. HG, applying the Mann–Whitney U test to the initial measurements.

The validation of the second hypothesis will be conducted through the application of the Kruskal–Wallis test to the vector of differences encompassing all groups, derived from the initial and final measurements and also by applying the Mann–Whitney U test with Bonferroni-corrected *p*-values.

Upon scrutiny of each group’s performance through the Mann–Whitney U test, as elucidated in Table 5 and Table 6, it is ascertained that the first hypothesis is true. According to the Mann–Whitney U test outcomes, the groups substantiate the same population at the commencement of the experimental period.

In Figure 13 and Figure 14, the vectors of differences between Week 0 and Week 5 are juxtaposed for all three groups, specifically focusing on unbalance time (%) detected on 2/3 axes and 3/3 axes. The comparative analysis reveals more pronounced differences when scrutinizing Group VG against Groups CG and HG, albeit the statistical significance level remains slightly below 20%. It is imperative to acknowledge the relatively modest strength of the second hypothesis, underscoring the necessity for caution, given the constrained 5-week testing interval, the limited sample size, and the fact that improvements have been detected for all groups. These constraints substantiate the recourse to non-parametric statistics. Future investigations warrant extending the study to encompass fencers from diverse geographical locations. Fencing is particularly niche in Romania, where the 30 individuals constituting the control and test groups represent approximately 20% of all fencers in the country with comparable experience.

A comprehensive examination of the outcomes in Table 7 and Table 8 shows that CG and HG exhibit comparable performances. However, discernible distinctions emerge in the performance of VG, albeit with a statistically low significance level when juxtaposed with the other groups.

## 4. Discussions

The present study introduces an augmented system featuring feedback modules, initially devised for the real-time monitoring of balance and movement control based on an MPU6500 accelerometer and a gyroscope sensor. In [38], an extensive analysis of the MPU6500 sensor’s reliability and performance is presented, revealing its widespread utilization as a low-cost and dependable sensor in various applications, including ground and aerial robotics. Notably, the gyroscope’s measurement error is demonstrated to be comparable to 10^−4^ [rad/s], which is one thousand times less than the designated reference threshold.

The extended system is tested to ascertain its efficacy in enhancing performance through real-time feedback modules, transcending its initial performance-monitoring role as outlined in [28]. Two distinct setups were employed to train two groups of fencers, while a control group served as a reference. One setup incorporated a visual feedback module, while the other used a haptic one.

All groups were randomly constituted and encompassed fencers boasting 4–5 years of experience. The visual feedback module incorporates an RGB LED light, dynamically signaling to fencers in real-time with a green hue when their movement aligns with a predefined reference value, as obtained from [28], and an orange indication when their movements surpass the reference value. In the second configuration, a smartwatch imparts haptic feedback through vibrations when fencers exceed the reference value.

The three groups underwent assessment initially at the commencement of the experiment and subsequently after a 5-week interval. In the control group and the groups employing the haptic feedback module, marginal enhancements were noted; however, statistical significance was not achieved according to the Wilcoxon test. The configuration manifesting tangible improvements is the monitoring system with visual feedback, demonstrating an average improvement exceeding 15%. Importantly, these results are substantiated by the Wilcoxon test, attesting to their statistical significance. In addition, Section 3.3 examines the vectors depicting the disparities between final measurements and initial measurements by applying the Kruskal–Wallis test and Mann–Whitney U test, with *p*-values adjusted using the Bonferroni correction. The outcomes of these supplementary analyses suggest that the likelihood of uniform improvements across all three groups, from a statistical perspective, is marginally below 20%. While this figure exceeds the conventional 5% threshold, it is imperative to consider the study’s limited training window of 5 weeks and the relatively small sample size of 10 subjects per group. Nonetheless, it is essential to underline that the initial test results, before the 5-week training window, were compared using the Mann–Whitney U test. The outcome was that the groups substantiated the same population at the commencement of the experimental period.

Additionally, as elucidated in the introduction, coaches routinely conduct analogous assessments of balance and movement control in their fencers through subjective observations [29]. The system’s objective evaluations were corroborated by the subjective assessments of ACS Floreta club coaches for 28 out of 30 fencers (93.33%) across both test groups and the control group, encompassing two sets of tests.

In future endeavors, developing and testing a new module centered on audio feedback is imperative. Notably, fencers are accustomed to deriving cues from their opponents’ visual and auditory signals to act or react efficiently. Simultaneously, comprehensive research endeavors are warranted, encompassing both novice fencers and those at the high-performance level.

The findings of this study carry significant implications for advancing technologies and training methodologies in the realm of sports by using IMUs. By showcasing the potential of the Internet of Things (IoT) and real-time sensorial feedback using IMUs, the study underscores the capacity of these innovations to augment performance in fencing. Typically, IMUs are employed to assess injury risks rather than enhance performance. This demonstration of efficacy may catalyze further exploration and innovation within sports technology and training methodologies.

### Strengths and Limitations

The main strengths of this study were that now, for the first time, there is a tool that can be used to improve balance and movement control in fencing based on real-time feedback, which demonstrated real impact for one setup that uses visual feedback clues. This tool works based on objectively quantifying a fencer’s balance and movement control performance by analyzing the torso angular velocity. However, there are some limitations to our analysis that should be noted. First, the number of fencers in the experimental groups and control group was only 10 per group. Notably, the entire population of 30 persons represents around 20% of all the fencers in Romania, which met the selection criteria from Section 2.4. Because of this, non-parametric statistics have been used to analyze the results’ statistical significance. Second, the test window was only five weeks long, limiting the expected improvements. The size of the test window has been selected to minimize absenteeism in the training of the test subject due to competitions, training camps, school exams, or the occurrence of some illnesses. It is important to note that the fencers have been selected from the same club to ensure that all subjects undergo similar training sessions.

## 5. Conclusions

The current investigation yields several noteworthy conclusions. Firstly, it substantiates the efficacy of the real-time monitoring system with visual feedback in enhancing balance and movement control among fencers. The recorded average improvement surpassing 15% proved statistically significant, as confirmed by the Wilcoxon signed-rank test. This outcome aligns with previous research highlighting the effectiveness of wearable devices employing inertial measurement units (IMUs) across diverse sports for optimizing training experiences.

Secondly, the study underscores the superiority of the visual feedback module over the haptic feedback module in augmenting performance. This observation suggests that for fencers, visual feedback may possess greater salience and informativeness in the context of refining balance and movement control. Nevertheless, additional research is imperative to validate this finding and explore potential synergies from integrating visual and haptic feedback within the real-time monitoring system.

Thirdly, it is crucial to acknowledge certain limitations in the study that may influence the interpretation of the results. The relatively modest sample size and the confined 5-week testing period are notable constraints. Furthermore, the study exclusively focused on fencing, warranting future research endeavors to replicate and broaden the generalizability of the findings.

In conclusion, exploring balance and movement control improvement in fencing through integrating the Internet of Things (IoT) and real-time sensorial feedback presents encouraging outcomes for performance enhancement. The real-time monitoring system, particularly with visual feedback, emerges as a potent tool for refining balance and movement control in fencers.

## Figures and Tables

**Figure 1 sensors-23-09801-f001:**
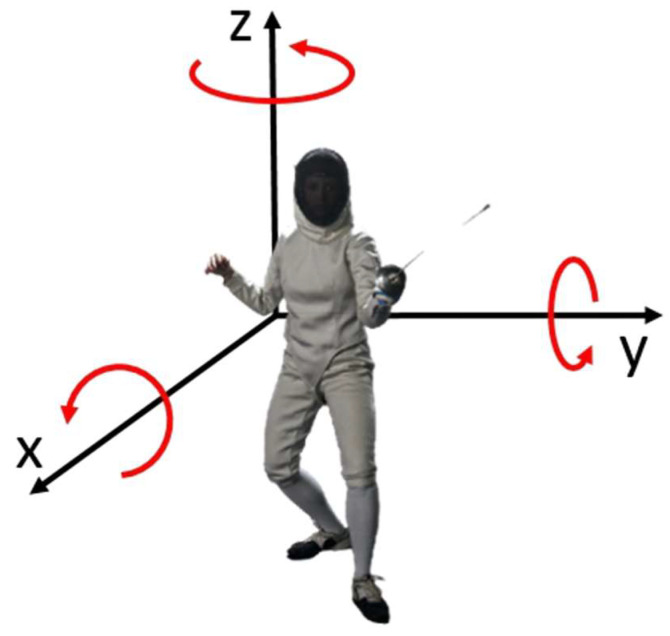
Movement on the fencing piste [28].

**Figure 2 sensors-23-09801-f002:**
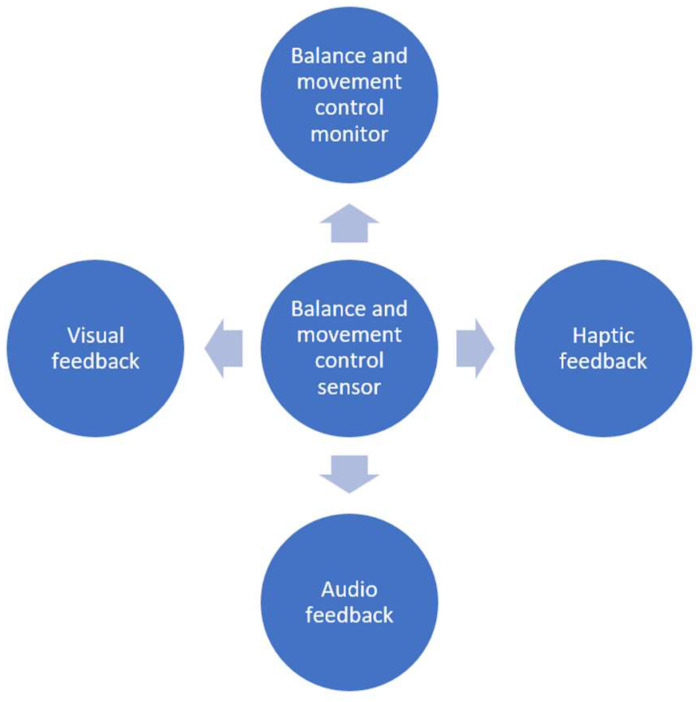
Main system blocks are used for enhancing balance and movement control in fencing.

**Figure 3 sensors-23-09801-f003:**
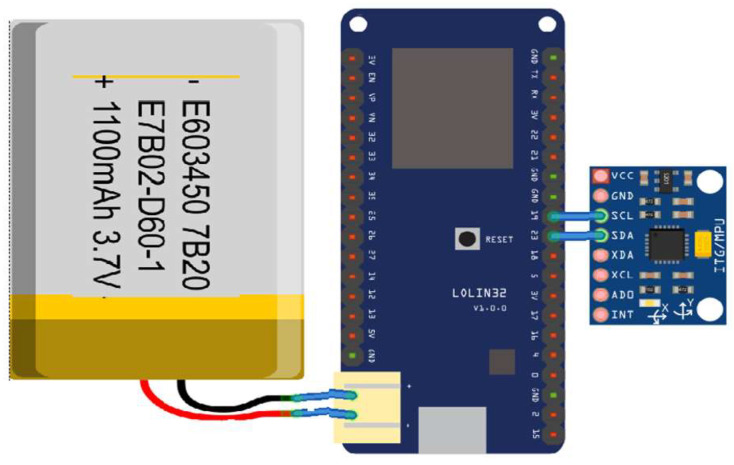
The balance sensor hardware [28].

**Figure 4 sensors-23-09801-f004:**
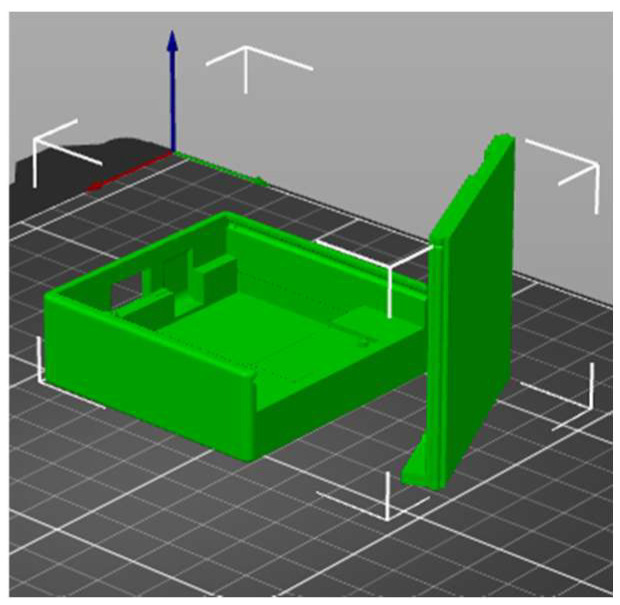
3D model of the sensor enclosure [28].

**Figure 5 sensors-23-09801-f005:**
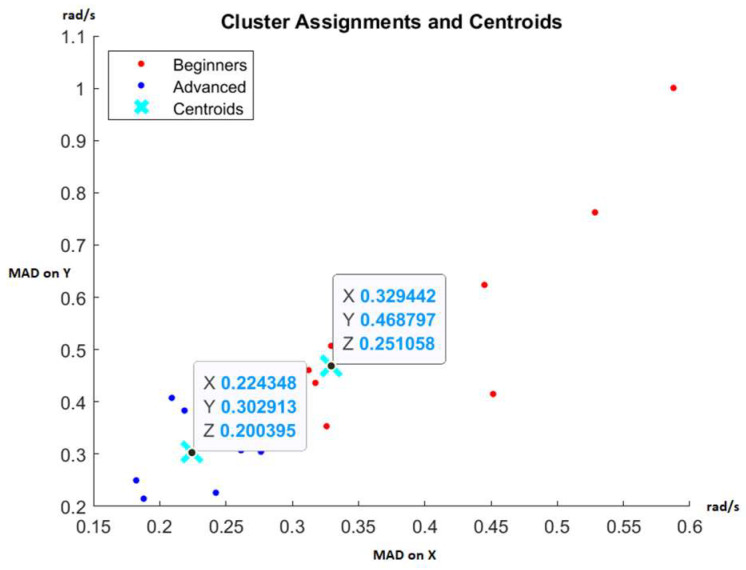
Cluster assignments using the K-means algorithm at 50% speed [28].

**Figure 6 sensors-23-09801-f006:**
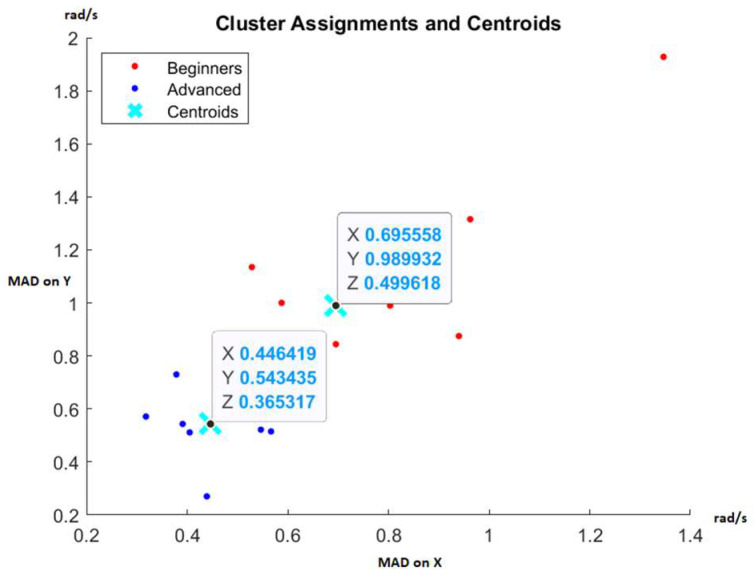
Cluster assignments using the K-means algorithm at 100% speed [28].

**Figure 7 sensors-23-09801-f007:**
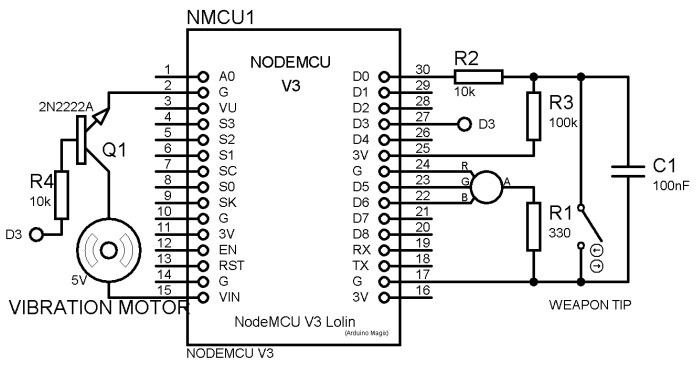
Haptic module schematic design [30].

**Figure 8 sensors-23-09801-f008:**
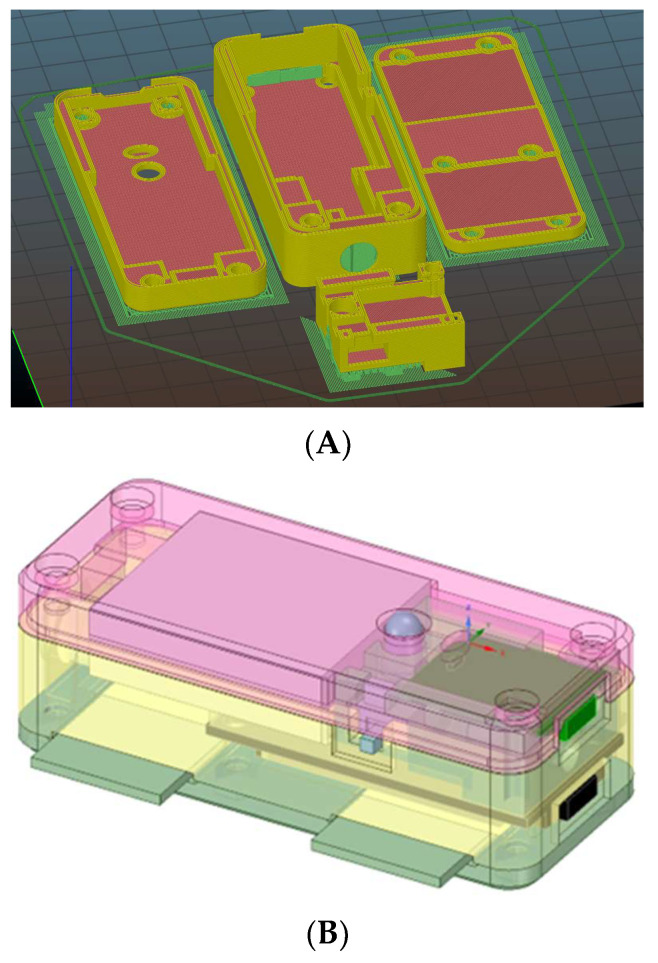
Haptic module 3D enclosure design. (**A**) Unfolded 3D case design based on four separated parts. (**B**) 3D model designed for 3D printing of the low-cost semi-scoring machine [30].

**Figure 9 sensors-23-09801-f009:**
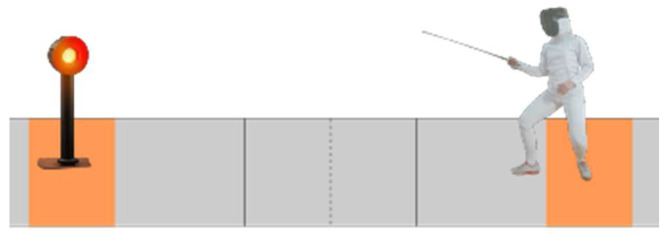
Visual feedback module setup.

**Figure 10 sensors-23-09801-f010:**
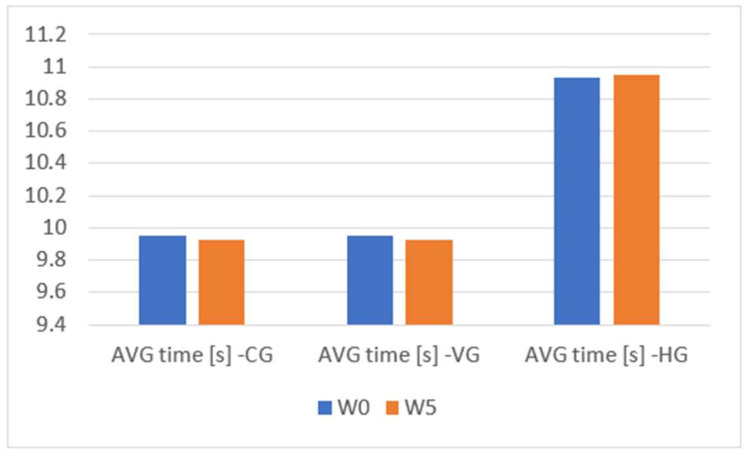
Average movement time comparison W0 vs. W5 for CG, VG, HG.

**Figure 11 sensors-23-09801-f011:**
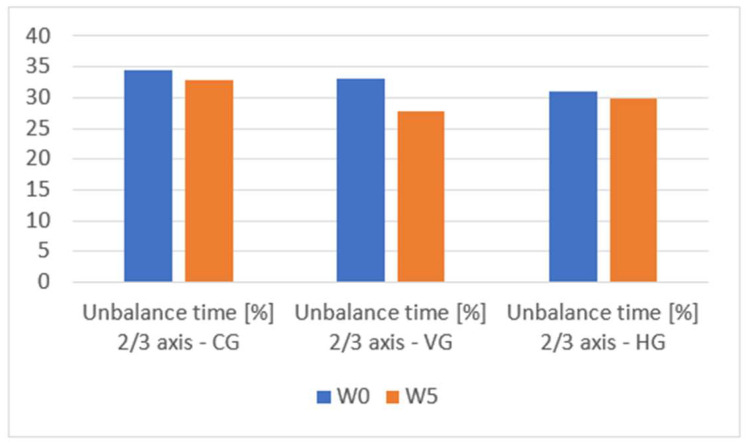
Unbalance time [%] 2/3 axis comparison W0 vs. W5 for CG, VG, HG.

**Figure 12 sensors-23-09801-f012:**
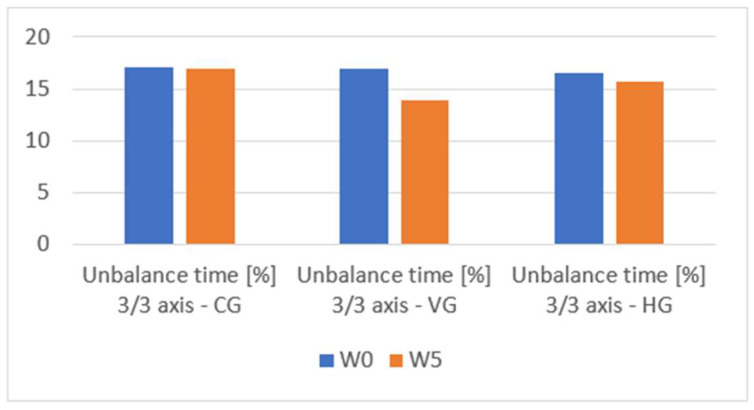
Unbalance time [%] 3/3 axis comparison W0 vs. W5 for CG, VG, HG.

**Figure 13 sensors-23-09801-f013:**
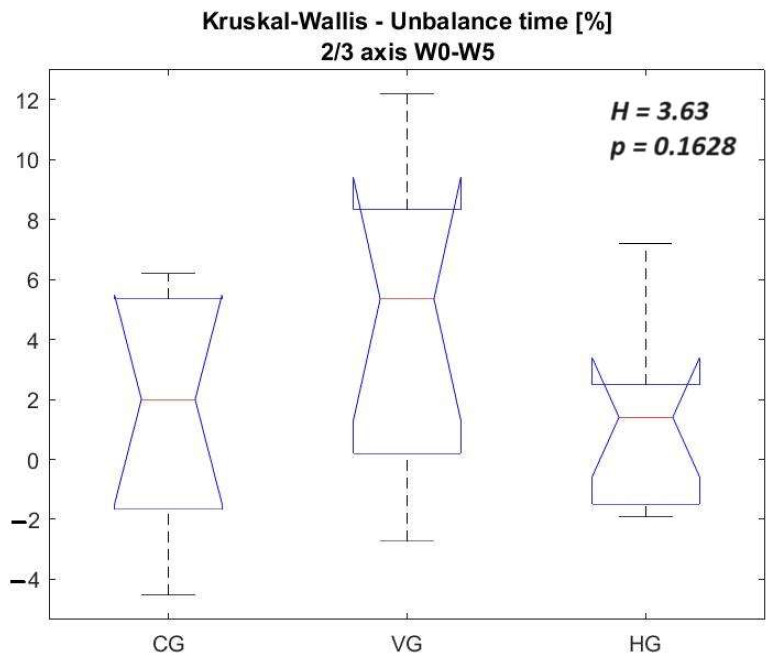
Kruskal–Wallis to the vector of differences between Week 0 and Week 5 for all groups—unbalance time [%] 2/3 axis.

**Figure 14 sensors-23-09801-f014:**
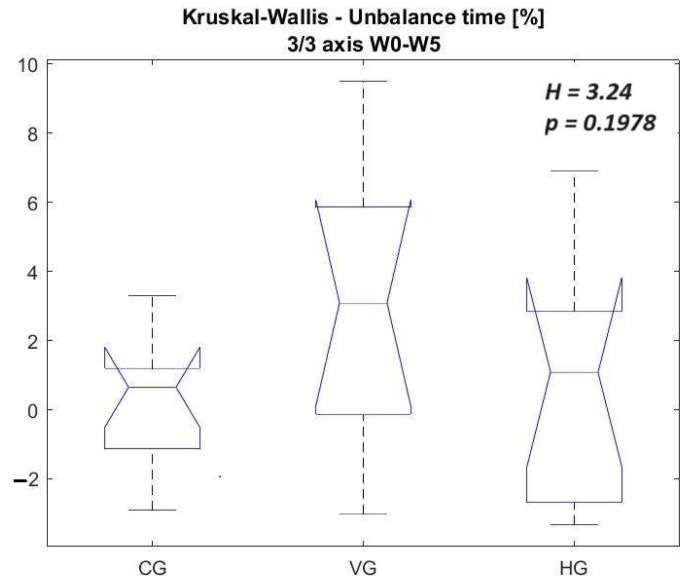
Kruskal–Wallis to the vector of differences between Week 0 and Week 5 for all groups—unbalance time [%] 3/3 axis.

**Table 1 sensors-23-09801-t001:** Reference thresholds for measuring balance and movement control in fencing.

X-Axis Threshold [rad/s]	Y-Axis Threshold [rad/s]	X-Axis Threshold [rad/s]
0.447	0.543	0.365

**Table 2 sensors-23-09801-t002:** Overall average comparison, Week 0 vs. Week 5.

Group	Week 0 Average Total Time [s]	Week 5 Average Total Time [s]	Week 0 Average Unbalance Time [%]—2/3	Week 5 Average Unbalance Time [%]—2/3	Week 0 Average Unbalance Time [%]—3/3	Week 5 Average Unbalance Time [%] 3/3
CG	9.952	9.928	34.366	32.775	17.152	16.905
VG	10.661	10.951	33.005	27.891	17.141	13.867
HG	10.933	10.948	31.028	29.816	16.58	15.766

**Table 3 sensors-23-09801-t003:** Relative comparison, Week 0 vs. Week 5.

Group	Week 0 vs. Week 5 Total Time [%]	Week 0 vs. Week 5 Unbalance—2/3 [%]	Week 0 vs. Week 5 Unbalance—2/3 [%]
CG	0.24174	4.629576	1.440065
VG	−2.64816	15.49462	19.1004
HG	−0.137011	3.906149	4.90953

**Table 4 sensors-23-09801-t004:** Wilcoxon signed-rank test for repeated measures results, Week 0 vs. Week 5.

Group/ Critical Value 8 *p* = 0.05	Wilcoxon Test Result W0 vs. W5 Unbalance—2/3 [%]	Wilcoxon Test Result W0 vs. W5 Unbalance—2/3 [%]
CG	15	22
VG	4	6
HG	14	21

**Table 5 sensors-23-09801-t005:** Mann–Whitney U test applied to unbalance time Week 0—2/3 axis.

	CG	VG
VG	0.7913	-
HG	0.5708	0.6232

**Table 6 sensors-23-09801-t006:** Mann–Whitney U test applied to unbalance time Week 0—3/3 axis.

	CG	VG
VG	1	-
HG	0.9097	0.9097

**Table 7 sensors-23-09801-t007:** Mann–Whitney U test, with Bonferroni-corrected *p*-values applied to vector difference of unbalance time Week 0—2/3 axis.

	CG	VG
VG	0.3239	-
HG	1	0.1513

**Table 8 sensors-23-09801-t008:** Mann–Whitney U test, with Bonferroni-corrected *p*-values applied to vector difference of unbalance time Week 0—3/3 axis.

	CG	VG
VG	0.2082	-
HG	1	0.3972

## Data Availability

The data presented in this study are available on request from the corresponding author.

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
