# Peer review of "Improving Balance and Movement Control in Fencing Using IoT and Real-Time Sensorial Feedback†"

_sensors, 2023, doi:10.3390/s23249801_

Round 1

Reviewer 1 Report

Comments and Suggestions for Authors

The manuscript presents a gyroscope-based system which measures the angular velocity of fencers’ torso around the three axes (X, Y and Z) while they are fencing, and then grants the possibility to provide fencers with feedback (either visually or haptically) to improve their balance and movement control. The authors first validated the ability of their measurement system (notably based on the mean absolute deviation) to ‘cluster’ beginners vs advanced fencers (using a K-means algorithm). They then coupled the system with visual and haptic feedback and measured whether providing these two types of feedback (notably as compared to a control group which did not receive any feedback) allows advanced fencers to improve their balance and movement control after five weeks of training. Significant improvements were only observed for the group which received visual feedback.

The paper is relatively well written and easy to read / understand. I think the presented results are interesting, but I have a couple of suggestions which I think would contribute to improve the paper.

First, whereas I think I understand how the haptic feedback was presented (vibrating watch), I am a bit confused about how the visual feedback was provided to the fencers. The authors talk about ‘real-time’ feedback, so I presume it was provided during the movement, but where was the LED ring located? In the mask? Somewhere else? This should be clearly explained in the paper, and more details should be given regarding the timing of the feedback (were there potential attention issues related to the feedback?).

In the results, the authors present long tables which I think are useful but a bit tedious to read. I would suggest including figures summarizing the main results. The tables could be provided as appendix / supplemental material.

Regarding the statistical analysis, the authors compared post and pre performance (week 5 vs week 0) for each group. Because this was done on repeated measures, I guess this was done using the Wilcoxon signed-rank test for repeated measures. The authors should specify this (currently they only wrote ‘Wilcoxon’, which could be confused with the rank-sum test). Along that line, I would suggest to also perform comparisons between groups. In particular, I would compare the vectors of differences (i.e., week5-week0) between the three groups using the Kruskal Wallis test. I would then perform paired comparisons between groups (i.e., CG vs VG, CG vs HG and VG vs HG) using the Wilcoxon rank-sum (Mann Whitney U) test, with Bonferroni-corrected p-values.

As a final note, I think the discussion is very short. The authors could notably discuss the fact that providing visual feedback significantly improved performance, whereas haptic feedback did not.

Comments on the Quality of English Language

none

Author Response

Dear Sir/Madam,

Thank you for the meticulous analysis of the proposed paper. All the recommendations you provided have improved the paper considerably.

The improvements, based on the feedback provided, are as follows:

Point 1:

The authors talk about ‘real-time’ feedback, so I presume it was provided during the movement, but where was the LED ring located? In the mask? Somewhere else? This should be clearly explained in the paper, and more details should be given regarding the timing of the feedback (were there potential attention issues related to the feedback?).

Response 1:

Thank you for pointing out that some information is missing regarding this module's description. Figure 9 has been added for clarity, line 256, which we described in lines 247-255.

Point 2:

In the results, the authors present long tables, which I think are useful but a bit tedious to read. I would suggest including figures summarizing the main results. The tables could be provided as appendix / supplemental material.

Response 2:

All the long tables have been moved to supplemental material and replaced with Figures 10-12 (lines 344-349), and the main results are centralized in tables 2 and 3 (lines 342, 343).

Point 3:

Regarding the statistical analysis, the authors compared post and pre performance (week 5 vs week 0) for each group. Because this was done on repeated measures, I guess this was done using the Wilcoxon signed-rank test for repeated measures. The authors should specify this (currently they only wrote ‘Wilcoxon’, which could be confused with the rank-sum test). Along that line, I would suggest to also perform comparisons between groups. In particular, I would compare the vectors of differences (i.e., week5-week0) between the three groups using the Kruskal Wallis test. I would then perform paired comparisons between groups (i.e., CG vs VG, CG vs HG and VG vs HG) using the Wilcoxon rank-sum (Mann Whitney U) test, with Bonferroni-corrected p-values.

Response 3:

Thank you for these recommendations. To solve this point, a new subsection has been introduced, 3.3. Lines 377-426.

Point 4:

As a final note, I think the discussion is very short. The authors could notably discuss that providing visual feedback significantly improved performance, whereas haptic feedback did not.

Response 4:

The discussion section has been extended, and a conclusion section has been added. Lines 428-439, 452-462, 473-512.

All the improvements regarding content have been highlighted. Also, the papers had gone through an English revision, but to keep more visible the content changes, the language improvements have not been highlighted. 

Reviewer 2 Report

Comments and Suggestions for Authors

Despite being an extension of a conference paper, the work presented does not show a high degree of originality and does not meet the high standards to be published in Sensors. The Materials and Methods section repeats what has already been reported in previously published conference papers. The initial part of the "Results" section should be moved to the Materials and Methods section, as it describes new tests done on three fencing groups (without access to any feedback, visual feedback, and haptic feedback). However, the "Results" section does not show a high presentation quality. The Discussion section is very poor. In this section, the results and their implications should be discussed more widely and thoroughly than the scientific landscape in the literature.

Author Response

Dear Sir/Madam,
Thank you for the analysis of the proposed paper. All the recommendations you provided have been taken into consideration.
The improvements, based on the feedback provided, are as follows:

Point 1:

Despite being an extension of a conference paper, the work presented does not show a high degree of originality and does not meet the high standards to be published in Sensors. 

Response 1:

We value your opinion, and we have to agree that from a technical point of view, the proposed system is simple. But, the practical implications are of high importance to fencing. The real-time monitoring system, particularly with visual feedback, emerges as a potent tool for refining balance and movement control in fencers. Also, after taking into account all the reviewers' recommendations, the paper has improved considerably. 

Point 2:

The Materials and Methods section repeats what has already been reported in previously published conference papers.

Response 2:

As the system used for monitoring balance and movement control in fencing is new, we considered presenting more details in the paper than just a standard citation. Afterward, we gave the system extension in Subsections 2.2 and 2.3. Also, regarding your recommendation, the initial description has been shortened, and an extra subsection, 2.4 Population and Sample lines 258-321 has been added.

Point 3:

The initial part of the "Results" section should be moved to the Materials and Methods section, as it describes new tests done on three fencing groups (without access to any feedback, visual feedback, and haptic feedback). However, the "Results" section does not show a high presentation quality. 

Response 3:

All the long tables have been moved to supplemental material and replaced with Figures 10-12 (lines 344-349), and the main results are centralized in tables 2 and 3 (lines 342, 343).

A new subsection has been introduced, 3.3. Vector difference comparison of the final and initial testing between the three groups using the Kruskal Wallis and Mann Whitney U tests, with Bonferroni-corrected p-values. Lines 377-426. 

Point 4:

The Discussion section is very poor. In this section, the results and their implications should be discussed more widely and thoroughly than the scientific landscape in the literature.

Response 4:

The discussion section has been extended, and a conclusion section has been added. Lines 428-439, 452-462, 473-512.

All the improvements regarding content have been highlighted. Also, the papers had gone through an English revision, but to be more visible the content changes the language improvements have not been highlighted. 

Reviewer 3 Report

Comments and Suggestions for Authors

The research aims in the abstract section: a new development system used for monitoring real-time balance and movement control of fencers based on a gyroscope that evaluates the angular velocity of the torso on the X, Y, and Z axes.

The authors of the research are requested to resolve the following points:

1)      There is no correspondence between the objective set out in the abstract section, the one at the end of the introductory section, and the first paragraph of the discussion section. The research objective must be the same in all sections. If there is no clarity in the proposed objective, a correct evaluation of the paper cannot be carried out.

2)      In the abstract section you need to place more relevant quantitative data.

3)      The second paragraph of the introductory section specifies the existence of instruments to measure indicators of sports performance in different sports (includes balance and movement control). It is necessary for the authors to specify in a paragraph the differences between the aforementioned instruments and the proposed new instrument.

4)      In the materials and methods section, a Population and Sample subsection is requested, taking into account the tests carried out on the athletes studied.

5)      In the materials and methods section, indicate the correlational statistics used, and the data normality test that justifies the use of non-parametric statistics.

6)      In the results section, there is content specific to the materials and methods section. Example: the formation of independent groups, the implementation methodology of the new instrument and the design of the testing scenario, among others.

7)      Since there are independent groups, it is recommended to use a statistic for two independent samples or for k independent samples according to SPSS (3 or more independent groups), given that the Wilcoxon test is for two related samples. The above will allow testing significant improvements or not in the feedback.

8)      In the penultimate paragraph of the discussion section, the idea is specified: “the coaches periodically do similar evalua-355 tions of balance and movement control of their fencers by subjective observations” (Line: 355-56). Demonstrate the idea veracity with at least one quote directly related to the research.

9)      It is recommended in the discussion section to cite some current paper on reliability indicators of measurement instruments, which the new instrument complies with.

10)   A section of conclusions, or final considerations, is requested. In this section, the most relevant scope of the research must be specified, which includes qualitative data and quantitative data.

I modestly consider that research is necessary for fencing, and has a lot of practical importance.

Comments on the Quality of English Language

Consult with an English language specialist

Author Response

Dear Sir/Madam,
Thank you for the meticulous analysis of the proposed paper. All the recommendations you provided have improved the paper considerably.
The improvements, based on the feedback provided, are as follows:

Point 1:

There is no correspondence between the objective set out in the abstract section, the one at the end of the introductory section, and the first paragraph of the discussion section. The research objective must be the same in all sections. If there is no clarity in the proposed objective, a correct evaluation of the paper cannot be carried out.

Response 1:

Thank you for observing the misalignment regarding the paper's objective in all the sections. There wasn't an obvious difference between the objective in the conference paper and the present proposed article. The extended system is designed to actively improve fencers’ capacities by incorporating real-time haptic and visual feedback mechanisms. The alignment has been made in the new paper abstract lines 12-25, end of introduction section lines 84-92, and discussion section lines 436-440.

Point 2:

In the abstract section you need to place more relevant quantitative data.

Response 2:

The abstract section has been revised. Lines 12-25.

Point 3:

The second paragraph of the introductory section specifies the existence of instruments to measure indicators of sports performance in different sports (includes balance and movement control). It is necessary for the authors to specify in a paragraph the differences between the aforementioned instruments and the proposed new instrument.

Response 3:

Usually, IMUs are used for assessing injury risks, as in [9-12] and [14], while the proposed system is used to evaluate balance and movement control performance. It is essential to mention that the proposed system considers the particular limitations of the movements made in fencing. Lines 47-50. 

Point 4:

In the materials and methods section, a Population and Sample subsection is requested, taking into account the tests carried out on the athletes studied.

Response 4:

An extra subsection, 2.4 Population and Sample lines 258-321, has been added.

Point 5:

In the materials and methods section, indicate the correlational statistics used, and the data normality test that justifies the use of non-parametric statistics.

Response 5:

Non-parametric tests were selected for these analyses due to the limited sample size. The objective was to monitor the statistical significance of the outcomes, considering the inherent constraints associated with the modest number of samples. Lines 318-321

Point 6:

In the results section, there is content specific to the materials and methods section. Example: the formation of independent groups, the implementation methodology of the new instrument and the design of the testing scenario, among others.

Response 6:

All the mentioned content has been moved to subsection 2.4 Population and Sample lines 258-321. Also, all the long tables have been moved to supplemental material and replaced with Figures 10-12 (lines 344-349), and the main results are centralized in tables 2 and 3 (lines 342, 343).

Point 7:

Since there are independent groups, it is recommended to use a statistic for two independent samples or for k independent samples according to SPSS (3 or more independent groups), given that the Wilcoxon test is for two related samples. The above will allow testing significant improvements or not in the feedback.

Resonse 7:

A new subsection has been introduced, 3.3. Vector difference comparison of the final and initial testing between the three groups using the Kruskal Wallis and Mann Whitney U tests, with Bonferroni-corrected p-values. Lines 377-426. 

Point 8:

In the penultimate paragraph of the discussion section, the idea is specified: “the coaches periodically do similar evalua-355 tions of balance and movement control of their fencers by subjective observations” (Line: 355-56). Demonstrate the idea veracity with at least one quote directly related to the research.

Response 8:

A quote has been introduced in the Introduction section, lines 59-63, to address the mentioned idea.

Point 9:

It is recommended in the discussion section to cite some current paper on reliability indicators of measurement instruments, which the new instrument complies with.

Response 9:

An extended analysis of the reliability of the used sensors is cited and shortly addressed in the paper at lines 428-435.

Point 10:

A section of conclusions, or final considerations, is requested. In this section, the most relevant scope of the research must be specified, which includes qualitative data and quantitative data.

Response 10:

The discussion section has been extended, and a conclusion section has been added. Lines 428-439, 452-462, 473-512.

All the improvements regarding content have been highlighted. Also, the papers had gone through an English revision, but to be more visible the content changes the language improvements have not been highlighted. 

Round 2

Reviewer 2 Report

Comments and Suggestions for Authors

I appreciated the revisions and changes made to the manuscript.  Now the manuscript has reached a better level. 

Reviewer 3 Report

Comments and Suggestions for Authors

The authors have improved the document as indicated